# NYSTRÖM SAMPLING DEPENDS ON THE EIGENSPECTRUM SHAPE OF THE DATA

**Djallel Bouneffouf**
IBM Thomas J. Watson Research Center
dbouneffouf@us.ibm.com

## ABSTRACT

Spectral clustering has shown a superior performance in analyzing the cluster structure. However, its computational complexity limits its application in analyzing large-scale data. To address this problem, many low-rank matrix approximating algorithms are proposed, including the Nyström method – an approach with proven approximate error bounds. There are several algorithms that provide recipes to construct Nyström approximations with variable accuracies and computing times. This paper proposes a scalable Nyström-based clustering algorithm with a new sampling procedure, Centroid Minimum Sum of Squared Similarities (CMS3), and a heuristic on when to use it. Our heuristic depends on the eigenspectrum shape of the dataset, and yields competitive low-rank approximations in test datasets compared to the other state-of-the-art methods.

## 1 INTRODUCTION

Spectral clustering techniques are widely used, due to their empirical performance advantages compared to other clustering methods Kong et al. (2011). However, a significant obstacle to scaling up spectral clustering to large datasets is that it requires building an affinity matrix between pairs of data points which becomes computationally prohibitive for large data-sets (Chen and Cai, 2011). To address this computational challenge, a common approach is to use the Nyström method as low-rank matrix approximation (Zhang and You, 2011), (Fowlkes et al., 2004). (Williams and Seeger, 2001). The method works by sampling a small set of landmark points from a large instances, to formulate an approximation for the eigen-decomposition of the full dataset using the sampled data. However, the performance of the approach is highly dependent on proper sub-sampling of the input data to include some *landmark points*, points that capture the inherent complexity and variability of the full dataset. Uniform sampling without replacement is the most used approach for this purpose (Fowlkes et al., 2004), (Cohen et al., 2014). Using properties of the data distribution a leading sampling algorithm has recently been suggested. The authors in (Bouneffouf and Birol, 2016), propose the ensemble minimum sum of the squared similarity sampling algorithm or ensemble-MS3. This algorithm is based on two works, the first one is the minimum sum of the squared similarity sampling or MS3 proposed in (Bouneffouf and Birol, 2015), that considers both the variance and the similarity of the dataset to select the landmark points. The second one is the ensemble Nyström methods proposed in (Kumar et al., 2009), which is a meta algorithm that combines the standard Nyström methods with the mixture weights. The ensemble-MS3 gives better results than the standard algorithms by increasing the accuracy compared with the standard Nyström method. However, the lack of speed is still a problem for the ensemble methods.

In this paper, we propose two algorithms that perform better than the ensemble MS3 and any existing ensemble Nyström algorithm. The first one, the "Centroid Minimum Sum of Squared Similarities algorithm" or CMS3 is an incremental sampling algorithm for Nyström based-spectral clustering. CMS3 improves the MS3 by adding centroid sampling upon the MS3, increasing the accuracy. In the first step, the algorithm starts sampling with a fixed number of initial landmark points and selects new landmark points one by one, such that the sum of the squared similarities between the previously selected points and the new point is minimized, and as a second step the algorithm selects only the centroid points from this sub-sample. The second one, the CMS3-tuned is deducted from the theoretical analyse of MS3 and leads to adapt the sampling according to the spectrum shape of the dataset.

## 2    CENTROID MINIMUM SUM OF SQUARED SIMILARITIES (CMS3)

The idea of the proposed algorithm CMS3 (described in Algorithm 1) is to sample $r$ points using MS3 where $m \leq r \leq X$ with the assumption that this sampling will give an $r$ convex points, and after that the CMS3 uses k-means (MacQueen et al., 1967) to cluster these $r$ points and select the centroids of these clusters as a global optimal landmark points. We could say that, the proposed algorithm is implemented under the following Hypothesis:

**Hypothesis 1.** *Comparing two similarity matrix $\widetilde{S}_m$ and $\widetilde{S}'_m$ corresponding to $CMS3$ and $MS3$ approximations, we have the following inequality between their error upper bounds:*

$$sup(||S - \widetilde{S}_m||) \leq sup(||S - \widetilde{S}'_m||)$$

---

**Algorithm 1** CMS3 Algorithm

---

1: **Input:** $X = \{x_1, x_2, ..., x_n\}$: dataset
  $m$: number of landmark data points
  $r$: number of landmark data points selected with MS3
  $\gamma$: size of the random subsampled set from the remaining data, in percentage
2: **Output:** $\widetilde{S} \in R^{m \times m}$: similarity matrix between landmark points
3: Initialize $\widetilde{S} = I_0$
4: $X_r := MS3(X, r, \gamma)$
5: $\widetilde{r} := kmeans(X_r, m)$
6: **For (i=0 to i$\leq$ m) do**
7:   $\widetilde{x}_i := \frac{1}{|\widetilde{r}_i|} \sum_{x_j \in \widetilde{r}_i} x_j$ //get centroid of the cluster $\widetilde{r}_i \in \widetilde{r}$
8:     $\widetilde{S} := \widetilde{S}_{\cup \widetilde{x}_i}$
9: **End For**

---

### 2.0.1    CMS3-TUNED:

Lemma 1 prescribes a method to select between $MS3$ and $CMS3$ methods.

**Lemma 1.** *Comparing the upper bound of $MS3$ and $CMS3$, as defined in Hypothesis 1. Assuming that $m\lambda_{m+1} + r\lambda_n << \lambda_2$, a necessary condition for $sup(||S - \widetilde{S}_m||) \leq sup(||S - \widetilde{S}'_m||)$ is $\lambda_2 \leq n\lambda_n$*

Following the Lemma 1 (the proof is in the appendix), the idea in the proposed algorithm (Algorithm 2), is to use $\lambda_2 \leq |sm| \times \lambda_{|sm|}$ as a switch condition for using CMS3 or MS3, where $|sm|$ is the sub-sampling size. These parameters could be seen as a proxy of the eigenspectrum shape of the data.

---

**Algorithm 2** CMS3-tuned Algorithm

---

1: **Input:** $X = \{x_1, x_2, ..., x_n\}$: dataset
  $m$: number of landmark data points
  $r$: number of landmark data points selected with MS3
  $\gamma$: size of the random subsampled set from the remaining data, in percentage
2: **Output:** $\widetilde{S} \in R^{m \times m}$: similarity matrix between landmark points
3:     $sm = Random(X, \gamma)$
4:     Compute $|sm|$ eigenvalues $\lambda_1, ..., \lambda_{|sm|}$ of the generalized eigenproblem $Pu = \lambda Du$; and let $Z \in R^{n \times |sm|}$ be the matrix containing the vectors $u_1, ..., u_{|sm|}$.
5:     if $|sm| \times \lambda_{|sm|} \geq \lambda_2$
6:     then $\widetilde{S} := CMS3(X, m, r, \gamma)$
7:     else $\widetilde{S} := MS3(X, m, \gamma)$

---

## 3    EVALUATION

We tested CMS3 and CMS3-tuned, and compared their performance to the results of four leading sampling methods described which are: Random sampling (RS), K-means sampling (KS) (Zhang et al., 2008), Minimum similarity sampling (SS) (Zeng et al., 2014), and Minimum sum of squared similarity sampling (MS3) (Bouneffouf and Birol, 2015). Notice that, we compare our algorithm

to ensemble Nyström rather than the standard Nyström, since it was shown earlier both in (Kumar et al., 2009) and in (Bouneffouf and Birol, 2016) that ensemble performs better than standard Nyström. We denote these algorithms as ensemble-RS, ensemble-KS, ensemble-SS, and ensemble-MS3, respectively.

We required each algorithm to sample 2%, 4%, 6%, 8% and 10% of the data as landmark points, which are used by Nyström-based spectral clustering methods to cluster the datasets. We have also tested the ensemble Nyström methods with different values of $p$, going from 2 to 10. Because sampling algorithms are sensitive to the datasets used, and clustering algorithms contain a degree of randomness, we used various benchmark datasets, and repeated our evaluations 1000 times. We measured the clustering quality of each algorithm using their average accuracy across these tests, also recording their standard deviations.

We compared the performance of the seven sampling methods using data from University of California, Irvine (UCI) Machine Learning Repository[1]. We chose nine datasets with different Instances, attributes and classes size: Abalone, Breast, Wine, Wdbc, Yeast, Shuttle, Letter, PenDigits and a7a.

Table 1 reports the average accuracy of each algorithm, along with their standard deviations across 1000 tests on the UCI datasets. As expected, the accuracies depend on the dataset. For example, the accuracy of all algorithms in Haberman problem and Wdbc datasets stay in the range of $50\%$, while going as high as over $89\%$ for the Abalone dataset. From this observation, we can say that the Haberman problem and Wdbc datasets present difficulties to Nyström method-based spectral clustering.

We note that, on these datasets, all tested algorithms have better performance than the baseline of random sampling. The results show that CMS3-tuned provided better clustering than the other algorithms on seven out of nine datasets, coming only narrowly second to CMS3 on the remaining two, though still within a standard deviation. Ranking the algorithms with respect to their mean accuracies, we note that the top two performing algorithms were CMS3-tuned and CMS3, in that order. The results on the UCI dataset confirm our heuristics that choosing between CMS3 and MS3 need to be done according to the spectrum shape of the dataset.

Table 1: Accuracy on UCI Datasets

| | Ensemble-SS | Ensemble-KS | Ensemble-RS | Ensemble-MS3 | CMS3 | CMS3-tuned |
|---|---|---|---|---|---|---|
| UCI Datasets | | | | | | |
| Abalone | $84.82 \pm 0.27$ | $85.74 \pm 0.31$ | $85.69 \pm 0.25$ | $86.44 \pm 0.40$ | $88.21 \pm 0.42$ | $\mathbf{89.19 \pm 0.21}$ |
| Breast | $67.85 \pm 0.33$ | $67.85 \pm 0.32$ | $67.83 \pm 0.34$ | $67.89 \pm 0.32$ | $68.94 \pm 1.17$ | $\mathbf{70.55 \pm 0.34}$ |
| Wine | $53.16 \pm 1.73$ | $55.17 \pm 3.80$ | $54.78 \pm 3.50$ | $67.99 \pm 3.67$ | $70.9 \pm 1.87$ | $\mathbf{71.39 \pm 2.01}$ |
| Wdbc | $51.32 \pm 0.13$ | $51.31 \pm 0.13$ | $51.30 \pm 0.12$ | $51.32 \pm 0.14$ | $\mathbf{52.98 \pm 0.07}$ | $52.31 \pm 0.13$ |
| Yeast | $67.58 \pm 0.13$ | $66.70 \pm 0.13$ | $66.92 \pm 0.12$ | $67.87 \pm 0.12$ | $69.06 \pm 0.07$ | $\mathbf{69.55 \pm 0.07}$ |
| Shuttle | $39.82 \pm 2.51$ | $37.90 \pm 2.89$ | $37.87 \pm 2.23$ | $41.45 \pm 3.85$ | $44.02 \pm 1.84$ | $\mathbf{44.31 \pm 1.94}$ |
| Letter | $41.77 \pm 1.83$ | $40.34 \pm 9.69$ | $38.66 \pm 9.77$ | $53.32 \pm 1.02$ | $56.44 \pm 3.70$ | $\mathbf{57.64 \pm 3.81}$ |
| PenDigits | $56.55 \pm 0.16$ | $56.46 \pm 0.21$ | $56.46 \pm 0.22$ | $56.94 \pm 0.19$ | $\mathbf{58.08 \pm 0.18}$ | $57.88 \pm 0.39$ |
| a7a | $16.45 \pm 1.17$ | $21.18 \pm 4.92$ | $22.06 \pm 4.08$ | $27.04 \pm 1.45$ | $25.89 \pm 3.14$ | $\mathbf{26.18 \pm 3.10}$ |

The results of the Ensemble-SS algorithm show overall better performance compared to Ensemble-KS and Ensemble-RS sampling. We also notice that the ensemble-MS3 gave higher performance than the sampling algorithms that are not based on MS3.

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

## A    KEY NOTION

This section focuses on introducing the key notions used in this paper.

### A.0.1    SPECTRAL CLUSTERING:

Spectral clustering algorithms employ the first $k$ eigenvectors of a Laplacian matrix to guide clustering. Loosely following the notation in Von Luxburg (2007), this can be outlined as follows. The algorithm takes as an input a number $k$ of clusters, an affinity matrix $S \in R^{n \times n}$ constructed using the cosine similarity between each pairs of data points, and as an output clusters $c_1, ..., c_k$. It starts by computing the Laplacian matrix $P = D - S$ ; where $D$ is an $n \times n$ diagonal matrix defined by $D_{ii} = \sum_{j=1}^{n} S_{ij}$, and after that it computes $k$ eigenvectors $u_1, ..., u_k$ corresponding to the first $k$ eigenvalues of the generalized eigenproblem $Pu = \lambda Du$; and let $Z \in R^{n \times k}$ be the matrix containing the vectors $u_1, ..., u_k$. Finally, it clusters $y_1, ..., y_n$ by k-means algorithm into clusters $c_1, ..., c_k$; with $y_i$ corresponding to the $i$-th row of $Z$. By analyzing the spectrum of the Laplacian matrix constructed over all data entries, the original data can be compressed into a smaller number of representative points using the Nyström approximation.

### A.0.2 Nyström Sampling:

If we consider $m$ landmark data points $L = l_1, l_2, ..., l_m$ from a given dataset $X = x_1, x_2, ..., x_n$ with $x_i \in R^n$ and $m \ll n$, then for any given point $x$ in $X$, Nyström method formulates

$$\frac{1}{m} \sum_{i=1}^{m} sim(x, l_i)\hat{\phi}(l_i) = \hat{\lambda}\hat{\phi}(x) \tag{1}$$

where $\hat{\phi}(x)$ is an approximation to the exact eigenfunction, $\hat{\lambda}$ is the corresponding approximate eigenvalue and $sim(x, y)$ denotes the similarity between $x$ and $y$. We can write the Eq.1 in matrix form, $\widetilde{S}\hat{\Phi} = m\hat{\Phi}\hat{\Lambda}$ where $\hat{\Phi} = [\hat{\phi}_1\hat{\phi}_2...\hat{\phi}_m]$ are the eigenvectors of $\widetilde{S}$ and $\hat{\Lambda} = diag\{\hat{\lambda}_1, \hat{\lambda}_2, ..., \hat{\lambda}_m\}$ is a diagonal matrix of the corresponding approximate eigenvalues. Then for an unsampled point $x$, the $j$-th eigenfunction at $x$ can be approximated by $\hat{\phi}_j(x) \simeq \frac{1}{m\hat{\lambda}_j} \sum_{i=1}^{m} sim(x, l_i)\hat{\phi}_j(l_i)$. With this equation, the eigenvector for any given point $x$ can be approximated through the eigenvectors of the landmark points $L$ Belabbas and Wolfe (2009). The same idea can be applied to approximate $k$ eigenvectors of $S$ by decomposing and then extending a $k \times k$ principal sub-matrix of $S$. First, let $S$ be partitioned as $S = [A B^\top B C]$ with $A \in R^{k \times k}$. Now, define spectral decompositions $S = U\Lambda U^T$ and $A = U_A \Lambda_A U_A^T$; the Nyström extension then provides an approximation for $k$ eigenvectors in $\widetilde{U} = [U_A B U_A \Lambda_A^{-1}]$ where the approximations of $\widetilde{U} \approx U$ and $\widetilde{\Lambda} \approx \Lambda$ may then be composed, yielding an Nyström approximation $\widetilde{S} \approx S$, with $\widetilde{S} = \widetilde{U}\Lambda_A \widetilde{U}^\top$. To measure the distance of these approximations, conventionally Frobenius norm is used.

### A.0.3 Minimum Sum of Squared Similarities

The MS3 algorithm Bouneffouf and Birol (2015) initially randomly chooses two points from the dataset $X$. It then computes the sum of similarities between the sampled points and a subset, $T$, selected randomly from the remaining data points. The point with the smallest sum of squared similarities is then picked as the next landmark data point. The procedure is repeated until a total of $m$ landmark points are picked.

---

**Algorithm 3** The Minimum Sum of Squared Similarities Algorithm

---

1: **Input:** $X = \{x_1, x_2, ..., x_n\}$: dataset
    $m$: number of landmark data points
    $\gamma$: size of the random sub-sampled set from the remaining data, in percentage
2: **Output:** $\widetilde{S} \in R^{m \times m}$: similarity matrix between landmark points
3: Initialize $\widetilde{S} = I_0$
4: **For (i=0 to i<2) do**
5:     $\widetilde{x}_i = Random(X)$
6:     $\widetilde{S} := \widetilde{S}_{\cup x_i}, \widetilde{X} := \widetilde{X} \cup \{\widetilde{x}_i\}$
7: **End For**
8: **While** $i < m$ **do**
9:     $T = Random(X\backslash\{\widetilde{X}\}, \gamma)$
10:     Find $\widetilde{x}_i = argmin_{x \in T} \sum_{j<i-1} sim^2(x, \widetilde{x}_j)$
11:     $\widetilde{S} := \widetilde{S}_{\cup \widetilde{x}_i}, \widetilde{X} := \widetilde{X} \cup \{\widetilde{x}_i\}$
12: **End While**

---

### A.0.4 Theoretical Study:

We propose here to study under which condition the proposed *Hypothesis 1* is valid. In order to do that, we propose at first to compute the the upper bound of the proposed sampling algorithm "CMS3" in *Theorem 1* and then compare it to the "MS3" upper bound in *Corollary 1*.

**Theorem 1.** *For a dataset $X = \{x_1, x_2, ..., x_n\}$, define the following positive definite similarity matrices, S: the $n \times n$ similarity matrix for the overall dataset with a maximum diagonal entry $S_{max}$, $\widetilde{S}_l$: a similarity matrix for $X_l$ with $l$ landmark point selected randomly from $X$, $\widetilde{S}_r$: a similarity matrix for $X_r$ with $r$ landmark point selected from $X_l$ using MS3, with $r \le l \le n$, $\widetilde{S}_m$: a similarity matrix for $X_m$ with $m$ landmark point selected from $X_r$ using K-means sampling, with $m \le r \le l \le n$; and $S_k$: the best possible rank-k approximation of*

*S. Then with some probability $1 - p$ or more, we can write*

$$||S - \widetilde{S}_m|| \quad \leq \quad 4T\sqrt{mC_X^{kern}Te} + mC_X^{kern}Te||W^{-1}|| + (r + 1)\sum_{i=r+1}^{n}\lambda_i + ||S - S_k|| \quad (2)$$

$$+ \quad nS_{\max}\sqrt[4]{\frac{64k}{l}}\left(1 + \sqrt{\frac{wd_S^*}{S_{\max}}}\right)^{\frac{1}{2}}$$

*where $||.||$ is the Frobenius norm, $d_S^* = \max_{ij}(S_{ii} + S_{jj}2S_{ij})$ and $w = -\frac{n-1}{2n-1}\frac{2}{\beta(l,n)}\log p$ with $\beta(l,n) = 1 - \frac{1}{2\max\{l,n-l\}}$*

*Proof.* Using the above notation, let us introduce some facts.

**Fact 1.** *Bouneffouf and Birol (2015) Let $\lambda_1 \geq ... \geq \lambda_n$ be the eigenvalues of the similarity matrix $S$, then with some probability $1 - p$ or more, we can write*

$$||S - \widetilde{S}_r|| \quad \leq \quad (r + 1)\sum_{i=r+1}^{n}\lambda_i + ||S - S_r|| + nS_{\max}\sqrt[4]{\frac{64k}{l}}\left(1 + \sqrt{\frac{wd_S^*}{S_{\max}}}\right)^{\frac{1}{2}}$$

**Property 1.** *Zhang et al. (2008) $(kern(a,b) - kern(c,d))^2 \leq C_X^{kern}(||a - c||^2 + ||d - b||^2), \forall a, b, c, d \in R$ where $C_X^{kern}$ is a constant depending on, the kernel $kern(.,.)$ and the sample set $X$.*

**Fact 2.** *Zhang et al. (2008) Let the whole sample set $X$ be partitioned into $g$ disjoint clusters $S_{kern}$, $c(i)$ being the function that maps each sample $x_i \in X$ to the closest landmark point $z_{c(i)} \in Z$. Then for some kernel $kern$ satisfying property (1), the partial approximation error $||S - \widetilde{S}_m||$ is bounded by*

$$||S - \widetilde{S}_m|| \leq 4T\sqrt{mC_X^{kern}Te} + mC_X^{kern}Te||W^{-1}|| \quad (3)$$

*where $T = max_{kern}|S_{kern}|$, and $e$ is the quantization error induced by coding each sample in $x_i \in X$ by the closest landmark point in $Z$, i.e., $e = \sum_{x_i \in X}||x_i - z_{c(i)}||^2$, and $||W^{-1}|| \in R^{m \times m}$ where $w_{ij} = k(z_i, z_j)$.*

By adding both sides of Eq.3 and Eq.3, noting that $\sum_{i=m+1}^{n}(.) \geq \sum_{i=r+1}^{n}(.)$ for positive argument and using the triangle inequality

$$||S - \widetilde{S}_m|| \leq ||S - \widetilde{S}_r|| + ||\widetilde{S}_r - \widetilde{S}_m|| \quad (4)$$

we prove Theorem 1. $\qquad\square$

**Corollary 1.** *The proposed Hypothesis 1 is valid, if and only if*

$$m \leq \frac{\lambda_r - r\sum_{i=r+1}^{n}\lambda_i - 4T\sqrt{(r-1)C_X^{kern}Te}}{C_X^{kern}Te||W^{-1}|| - \sum_{i=r}^{n}\lambda_i} \quad (5)$$

*Proof.* Assuming the comparison of the upper bounds appears with the inequality,

$$4T\sqrt{mC_X^{kern}Te} + mC_X^{kern}Te||W^{-1}|| + (r + 1)\sum_{i=r+1}^{n}\lambda_i + ||S - S_k|| + nS_{\max}\sqrt[4]{\frac{64k}{l}}\left(1 + \sqrt{\frac{wd_S^*}{S_{\max}}}\right)^{\frac{1}{2}}$$

$$\leq (m + 1)\sum_{i=m+1}^{n}\lambda_i + ||S - S_k|| + nS_{\max}\sqrt[4]{\frac{64k}{l}}\left(1 + \sqrt{\frac{wd_S^*}{S_{\max}}}\right)^{\frac{1}{2}} \quad (6)$$

after simplification we get

$$4T\sqrt{mC_X^{kern}Te} + mC_X^{kern}Te||W^{-1}|| \quad \leq (m + 1)\sum_{i=m+1}^{n}\lambda_i - (r + 1)\sum_{i=r+1}^{n}\lambda_i \quad (7)$$

Knowing that $m \leq r$ we can write

$$4T\sqrt{mC_X^{kern}Te} + mC_X^{kern}Te||W^{-1}|| \quad \leq (m-r)\sum_{i=r+1}^{n}\lambda_i + (m+1)\sum_{i=m+1}^{r}\lambda_i$$

then replacing $m$ by $r-1$ gives,

$$m \leq \frac{\lambda_r - r\sum_{i=r+1}^{n}\lambda_i - 4T\sqrt{(r-1)C_X^{kern}Te}}{C_X^{kern}Te||W^{-1}|| - \sum_{i=r}^{n}\lambda_i} \tag{8}$$

We note that going from inequality (8) back to (6) is straightforward, and can be achieved by tracing the above steps in reverse. $\square$

### A.0.5   CMS3-TUNED:

We propose here to use the above theoretical results to propose an improved version of the CMS3. Corollary 1 prescribes a method to select between $MS3$ and $CMS3$ methods. However, due to its complexity, the idea here is to relax the "if and only if" of the Corollary 1 as follows:

**Corollary 2.** *Comparing the upper bound of $MS3$ and $CMS3$, as defined in Hypothesis 1. Assuming that $m\lambda_{m+1} + r\lambda_n << \lambda_2$, a necessary condition for $sup(||S - \widetilde{S}_m||) \leq sup(||S - \widetilde{S}'_m||)$ is $\lambda_2 \leq n\lambda_n$*

*Proof.* From Eq. (8) a necessary condition for having the Corollary 1 could be the following:

$$0 \quad \leq (m-r)\sum_{i=r+1}^{n}\lambda_i + (m+1)\sum_{i=m+1}^{r}\lambda_i$$

then the following still hold,

$$0 \quad \leq (m-r)(n-r)\lambda_n + (m+1)(r-m)\lambda_{m+1}$$

which implies

$$0 \quad \leq (m+1)\lambda_{m+1} - (n-r)\lambda_n$$

with $\lambda_{m+1} \leq \lambda_2$ and assuming that $m\lambda_{m+1} + r\lambda_n << \lambda_2$, gives us $\lambda_2 \leq n\lambda_n$ $\square$

Following the Corollary 2, the idea in the proposed algorithm (Algorithm 2), is to use $\lambda_2 \leq |sm| \times \lambda_{|sm|}$ as a switch condition for using CMS3 or MS3, where $|sm|$ is the sub-sampling size. These parameters could be seen as a proxy of the eigenspectrum shape of the data.

