# OpenReview forum: "NYSTROM SAMPLING DEPENDS ON THE EIGENSPECTRUM SHAPE OF THE DATA"
_ICLR.cc/2018/Workshop — Reject_

### Official Review · AnonReviewer3 · 2018-03-02
**Not convincing and only barely relevant**

**Rating:** 5
**Confidence:** 4

**Review:**

The method attacks the scalability of spectral clustering, but all experimented data sets are not big enough to show the benefits. The full spectral clustering method can be applied to all these data sets.

Moreover, save computational cost in a clustering method, even if it is correct, is not so relevant to ICLR.

---

### Official Review · AnonReviewer2 · 2018-03-09
**new sampling technique for Nystrom approximation**

**Rating:** 5
**Confidence:** 4

**Review:**

The authors consider modifying a prior diversity-based (MS3) landmark sampling scheme for forming Nystrom approximations by oversampling landmarks using MS3, clustering with kmeans, then using the centroids of the clusters as the final landmark points to form a Nystrom approximation (they call this CM3). They further argue that CM3 is more accurate than MS3 only if the smallest eigenvalue of the PSD matrix is sufficiently large compared to its second eigenvalue, and they argue that this relationship can be checked approximately with a heuristic based on a random subsample of the PSD matrix.

The idea of clustering to select landmark points is not novel in the area, so the novelty lies in the application to MS3 and their suggested tuning method to select between CM3 and M3. Weighing the pros and cons, I suggest not accepting this paper; my major considerations in reaching that decision are the lack of compelling evidence that CM3 is useful across a wider range of kernels and the poor quality of the writing.

Pros:
- Empirically, they show their algorithm outperforms standard ensemble Nystrom methods when a comparable number of landmark points are sampled/created.

Cons:
- As it is, the only reasonable claim supported by their experiments is that CM3 outperforms other Nystrom approaches specifically in the case of the cosine similarity kernel. That is because the cosine similarity kernel is used in all experiments. However,  this is not surprising, as the diversity measure used in MS3 sampling is based on the cosine distance. It would be far more meaninful/insightful/useful to see a comparison of the performance on other kernels, especially the most used kernel in practice, the Gaussian kernel.

- Only the final number of landmark points are constrained to be the same across the algorithms in the experiments. Thus one can argue that CM3 is more accurate with respect to spectral clustering using a fixed rank approximation. However, it is also useful to know how many overall samples of the similarity matrix had to be computed (we want to avoid sampling a large fraction) for all the algorithm. This information is not made available: for instance, the CM3-tuned algorithm is presumably sampling more of the similarity matrix than all the other algorithms since it computes the eigenspectrum of a subsample of the similarity matrix to decide whether to use CM3 or M3, and is sampling from a pool of random samples when it does the M3 stage.

- The paper is heavy on parameters, and they are often used before they are defined, and in conflicting ways (e.g. at several points sets are used as numbers). Hypothesis 1 is confusing: the norm should be specified, and what is sup?

- The theory is impenetrable despite looking very straight-forward; this is due to the poor notation and terseness of the description of the parameters and how the bounds from previous work are put together to obtain their results.

- Similarly, the description of Nystrom sampling is unnecessarily mystifying (e.g., the partitioning of S is incorrect; presumably because of poor formatting) to ML audiences. See e.g. section 2.1 of "Making Large-Scale Nystrom Approximation Possible" by Li et al. for a more accessible linear-algebraic description of Nystrom.

---

### Official Review · AnonReviewer1 · 2018-03-11
**A new sampling strategy for Nystrom**

**Rating:** 3
**Confidence:** 5

**Review:**

Summary:
The paper proposes a new sampling strategy for Nystrom's method which the author calls CMS3. The algorithm slightly modifies MS3, which is an existing algorithm that performs the same task.
CMS3 clusters r points obtained using MS3 to obtain m cluster centroids and returns the approximated similarity matrix based on these m clusters centroids.
Another version of the algorithm, called CMS-Tuned, has also been proposed. CMS-Tuned adapts to the data to switch between CMS3 and MS3.
Theoretical justification has been provided for choosing between CMS3 and MS3 in CMS3 tuned.

Issues:
The paper has not been written clearly, the notation is confusing. Symbols have been referred before their definition – for example, P is algorithm (2).
Because of confusing notation, it is hard to understand the proofs. Most probably they are not correct. For example, consider equation (3), that has been supposedly obtained from equation (2) and fact (1). Since equation (2) and fact (1) are both <= type inequality, it is not clear how they can be used to arrive at equation (3)
Obtained results are not that good as compared to existing methods.
Since CMS3-Tuned performs better than MS3-Tuned in general, on an average, selecting MS3 is more beneficial than CMS3, then why do we need CMS3 in the first place?

Overall Opinion: Reject, since the paper has not been written clearly, and even if everything is correct, it does not help in any major way.

---

### Decision · Program_Chairs · 2018-03-20
**ICLR 2018 Workshop Acceptance Decision**

**Decision:**

Reject

**Comment:**

Based on the reviews, this paper has not been accepted for presentation at the ICLR workshop. However, the conversation and updates can continue to appear here on OpenReview.